# Loss of ninein interferes with osteoclast formation and causes premature ossification

Thierry Gilbert[1], Camille Gorlt[1,2†], Merlin Barbier[1†], Benjamin Duployer[3†], Marianna Plozza[2†], Ophélie Dufrancais[2], Laure-Elene Martet[1], Elisa Dalbard[1], Loelia Segot[1], Christophe Tenailleau[3], Laurence Haren[1], Christel Vérollet[2,4], Christiane Bierkamp[1], Andreas Merdes[1]*

[1]Molecular, Cellular and Developmental Biology, Centre de Biologie Intégrative, UMR5077, CNRS & Université Paul Sabatier, Toulouse, France; [2]Institut de Pharmacologie et de Biologie Structurale, UMR5089, CNRS & Université Paul Sabatier, Toulouse, France; [3]CIRIMAT, UMR5085, CNRS & Université Paul Sabatier, Toulouse, France; [4]International Research Project CNRS "MAC-TB/HIV", Toulouse, France

**\*For correspondence:**
andreas.merdes@univ-tlse3.fr

[†]These authors contributed equally to this work

**Competing interest:** The authors declare that no competing interests exist.

**Abstract** Ninein is a centrosome protein that has been implicated in microtubule anchorage and centrosome cohesion. Mutations in the human *NINEIN* gene have been linked to Seckel syndrome and to a rare form of skeletal dysplasia. However, the role of ninein in skeletal development remains unknown. Here, we describe a ninein knockout mouse with advanced endochondral ossification during embryonic development. Although the long bones maintain a regular size, the absence of ninein delays the formation of the bone marrow cavity in the prenatal tibia. Likewise, intramembranous ossification in the skull is more developed, leading to a premature closure of the interfrontal suture. We demonstrate that ninein is strongly expressed in osteoclasts of control mice, and that its absence reduces the fusion of precursor cells into syncytial osteoclasts, whereas the number of osteoblasts remains unaffected. As a consequence, ninein-deficient osteoclasts have a reduced capacity to resorb bone. At the cellular level, the absence of ninein interferes with centrosomal microtubule organization, reduces centrosome cohesion, and provokes the loss of centrosome clustering in multinucleated mature osteoclasts. We propose that centrosomal ninein is important for osteoclast fusion, to enable a functional balance between bone-forming osteoblasts and bone-resorbing osteoclasts during skeletal development.

## Editor's evaluation

This valuable study offers new insight into the role of centrosome protein ninein in skeletal development through an analysis of the skeletal phenotype of ninein-deficient mice. The evidence supporting the conclusion is convincing. This work will be of interest to scientists in bone biology and skeletal development field.

## Introduction

The centrosome is a small organelle present in most animal cells. It is composed of two cylindrically shaped centrioles that are surrounded by proteins of the pericentriolar material (*Prigent and Uzbekov, 2022*). Because the pericentriolar material contains large amounts of gamma-tubulin-complexes necessary for microtubule nucleation, the centrosome is generally viewed as a major

microtubule-organizing center (**Moritz et al., 1995**). The two centriolar cylinders can be distinguished morphologically: the older one, the mother centriole, carries specific structures termed distal and subdistal appendages. Whereas distal appendages contribute to ciliogenesis, subdistal appendages are involved in microtubule anchorage (**Prigent and Uzbekov, 2022**). This anchorage results in a microtubule network that is radially organized from the centrosome, as seen in undifferentiated cells or in fibroblasts.

Mutations of multiple genes that encode centrosomal proteins are linked to developmental disorders such as autosomal recessive primary microcephaly (MCPH; **Thornton and Woods, 2009**), microcephalic primordial dwarfism (MPD; **Chavali et al., 2014**), and ciliopathies (**Rao Damerla et al., 2014**). MCPH is characterized by reduced head circumference at birth with a decreased size of the cerebral cortex and cognitive defects. Many microcephaly-associated genes are implicated in centrosome formation, or in the assembly and orientation of mitotic spindles (**Gabriel et al., 2020**; **Pirozzi et al., 2018**). So far, abnormal cortical brain development is viewed as the principal cause of microcephaly. In addition to the microcephalic phenotype, features such as growth retardation, malformed limbs, or cranial defects are found in MCPH-related disorders like Seckel syndrome, Meier-Gorlin syndrome, and microcephalic osteodysplastic primordial dwarfism (**Klingseisen and Jackson, 2011**). As in MCPH, a common hallmark of these syndromes are mutations affecting the integrity of centrosomes.

Among the genetic loci for Seckel syndrome, ninein was characterized more recently (**Dauber et al., 2012**). Ninein is an evolutionarily conserved component that binds to the subdistal appendages at the mother centriole and at the basal body of the primary cilium, and to the proximal ends of both mother and daughter centrioles. At the cellular level, it has been proposed that ninein anchors gamma-tubulin complexes and microtubules to centrosomal and non-centrosomal sites (**Mogensen et al., 2000**; **Dammermann and Merdes, 2002**; **Delgehyr et al., 2005**), and that it is necessary for the integrity of mitotic spindle poles and for spindle orientation (**Logarinho et al., 2012**; **Lecland et al., 2019**). Moreover, ninein has been described as a regulator of cell migration, as it controls the dynein/dynactin-dependent release of microtubules from the centrosome, which is believed to stabilize lamellopodial extensions during cellular movement (**Abal et al., 2002**).

At the level of the organism, ninein was found to be essential for early brain morphogenesis in zebrafish (**Dauber et al., 2012**). This is consistent with a proposed role of ninein in maintaining neural progenitor cells in the developing mammalian neocortex (**Wang et al., 2009**; **Shinohara et al., 2013**). Another report identified mutations in ninein as causative for spondyloepimetaphyseal dysplasia with joint laxity (**Grosch et al., 2013**). The hallmarks of this disorder include short stature, midface hypoplasia, joint laxity with dislocations of the hip and knee joints, genua valga, progressive scoliosis and long, slender, fingers. Although this pathogenesis is not well understood, an involvement of ninein in skeletal development was proposed (**Grosch et al., 2013**). Skeletal development involves ossification of the skull and of long bones. Whereas skull bone is formed by intramembranous ossification, long bones develop by replacement of cartilage templates, previously established by chondroblasts. Both ossification pathways include the action of specific cell types, mainly involving osteoblasts and osteoclasts. Osteoblasts are derived from mesenchymal stem cells and produce bone mineral in the form of calcium phosphate crystals. This action is counterbalanced by mineral-resorbing osteoclasts, to control bone growth and homeostasis (**Boyle et al., 2003**; **Phan et al., 2004**). Bone formation and resorption is a coupled process, illustrated by the fact that osteoclast differentiation and function depend on paracrine stimulation by osteoblasts (**Phan et al., 2004**). The differentiation of osteoclasts occurs from myeloid precursor cells and involves cell fusion into large syncytial cells. The formation of large osteoclasts appears to enhance their bone resorption activity, as smaller syncytia have been found to be less effective (**Lees and Heersche, 1999**; **Møller et al., 2020**; **Dufrançais et al., 2021**).

To characterize the role of ninein in bone formation, we investigated ninein-deficient mice during early development. Previous work on ninein-knockout mice had revealed mild defects in the epidermal barrier, without obvious abnormalities in other tissues (**Lecland et al., 2019**). In the present study, we analyzed these mice in more detail and detected temporary but strong irregularities during skeletal development, leading to premature endochondral and intramembranous ossification. Importantly, we found that fusion of precursor cells into mature and functional osteoclasts is impaired in ninein-deficient mice.

## Results

### Ninein-deficiency and in-utero-development

Constitutive ninein-knockout mice were viable, without obvious abnormalities, and capable of reproduction (*Lecland et al., 2019*). However, when ninein-deficient animals were intercrossed, a significant reduction in the litter size was observed (*Figure 1A*). Following delivery, nearly one third of the newborn pups were severely growth-retarded with signs of dwarfism, reminiscent of what is occurring in patients carrying ninein mutations (*Dauber et al., 2012*; *Grosch et al., 2013*). Most growth-retarded pups identified at birth did not survive, and the neonatal lethality was reaching 35% within 48 hr. Whereas spontaneous delivery occurred generally on the nineteenth day post coitum (dpc) in control females, several ninein-deleted females had difficulty to deliver by the 22nd dpc and were then euthanized. Examination of the litters revealed that they never contained more than four pups and that these pups were always dead (*Figure 1B*). This increased mortality in utero was exclusively dependent on the zygotic genotype, as ninein knock-out females following mating with wild type males displayed normal litter size and no growth retardation among pups (*Figure 1A*). Examination of the uterine horns at mid-gestation, at embryonic day 9.5 post-coitum (E9.5dpc), revealed a similar number of deciduae in wild type and ninein-deleted mothers (8.7±0.8, n=13 vs 9.2±0.8, n=12, respectively). However, following removal of the extraembryonic tissues, very small ninein del/del embryos were frequently found, as well as empty decidua (*Figure 1C*). At E11.5, extremely small embryos were occasionally observed (*Figure 1D*). A significant reduction in the progeny occurred at mid gestation and accounted for the reduced number of pups observed at birth. The potential role of ninein in these viability-related processes in utero will not be addressed here. In the following experiments, mating between heterozygous mice (del/+) was performed, yielding regular litter sizes. This confirmed that prenatal loss of embryos was largely dependent on the del/del zygotic genotype. Moreover, heterozygous intercrosses enabled quantitative morphological comparison between del/+ (control) and homozygous del/del siblings at the same embryonic stage.

### Early endochondral ossification in ninein-deleted mice

Because ninein has been implicated in the formation of primary cilia (*Graser et al., 2007*), we looked whether ciliopathy phenotypes were observed in embryos lacking this centrosomal protein. We first analyzed embryos on day 18.5 of gestation, as most organs were sufficiently differentiated to assess the presence of developmental defects. No external abnormalities were noticed and polydactyly was never observed upon ninein deprivation. Visceral organs such as the liver and kidney were free of cysts, in contrast to mice mutant for other proteins of centrosomes and cilia, such as Cep290 or Cc2d2a (*Rachel et al., 2015*; *Veleri et al., 2014*). Neither laterality defects nor airway defects were present in ninein del/del mice. No morphological abnormalities were noticed in sensory organs such as the eye and ear, although these were not tested functionally. Despite similar embryo sizes of control and knockout mice, attention was paid to the course of skeletal development, as bone dysplasia was reported in patients carrying a ninein mutation (*Grosch et al., 2013*). Whole embryos were stained with Alcian blue and Alizarin red, to investigate the formation of cartilage and bone, respectively (*Figure 2A*). Examination of E18.5 embryos revealed an advanced ossification in the digits of all ninein-deficient embryos. In the forelimb of control embryos, one ossification center was detected in the metacarpal bone and in the proximal phalange of digits 2–5 (*Figure 2B*, left panel). By contrast, in mutant embryos, an additional mineralized area was clearly visible in the intermediate phalange of digits 2–4 (*Figure 2B*, right panel). A similar advanced ossification was observed in the hindlimbs of ninein-deprived embryos (*Figure 2C*, right image). The proximal phalange of digit one exhibited a small ossification center, which was never observed in control embryos at this stage. The intermediate phalanges of digits 2–4 started to mineralize in ninein del/del embryos, but not in controls, and the percentage of digits containing three ossification centers increased (graphs below each image in *Figure 2B and C*). The data indicate that early endochondral ossification occurred in embryos lacking ninein. However, no differences in bone size were seen in the two groups (*Figure 2D*, and morphometric analysis in *Supplementary file 1*). We then looked at the neck region with emphasis on the first and second cervical vertebrae. The atlas exhibited the same degree of ossification in all pups. On the opposite, the second vertebra, the axis, exhibited an additional ossification center within the cartilaginous vertebral body in ninein-deleted embryos (*Figure 2E*). To further characterize the onset of this earlier ossification in the limbs, we examined bone formation in controls and mutants at

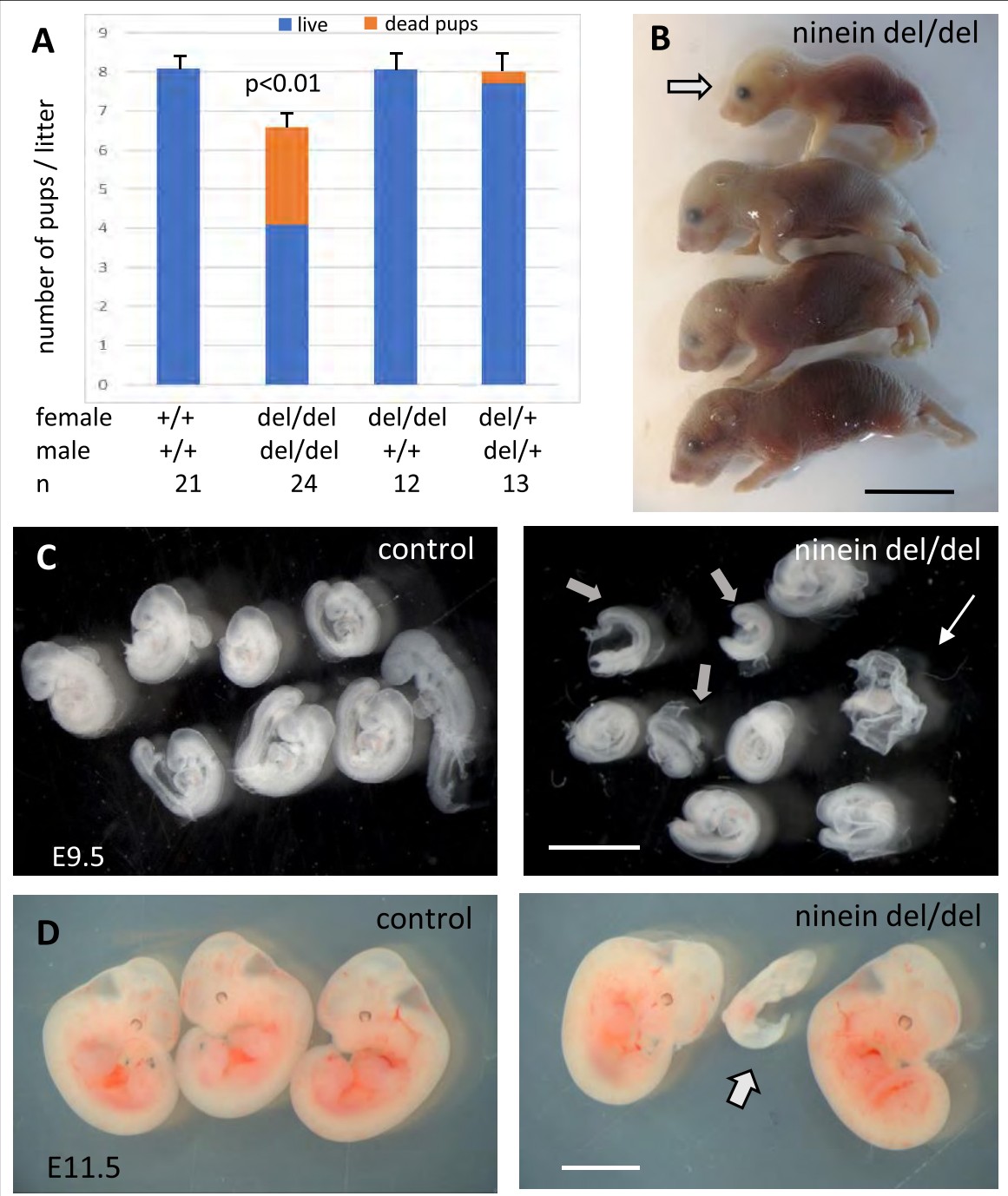

**Figure 1.** Ninein knock-out mice display reduced litter size upon a zygotic genotype. (**A**) Litter size comparisons following crossings among control or ninein-deleted animals at birth. Blue-colored bars: live pups, orange-colored bars: dead pups. Crossing of males and females with homozygous deletion of ninein leads to systemic prenatal death as compared to control matings. The number of pups alive is reduced by nearly 50%. (**B**) Example of a litter with dead newborns, one of them being of very small size (arrow) is shown. (**C, D**) Examination of deciduae at mid-gestation, at E9.5 and E11.5, in control and ninein del/del embryos. Despite similar numbers of deciduae in both groups, some dissected deciduae of ninein del/del contained only small fetuses (arrows in **C, D**) or no fetus (thin arrow in **C**). Bars, (**B**) 1 cm, (**C**) 2 mm, (**D**) 5 mm.

E16.0 and E16.5. First, E16 embryos were stained with alizarin red, to visualize the mineralized bones only (***Figure 3A***). In controls, no ossification was visible in the upper and lower foot. In ninein del/del embryos, two metacarpal and three metatarsal bones showed signs of mineralization (***Figure 3A***, arrows). In slightly older embryos (E16.5), control metacarpal bones 3 and 4 exhibited tiny red deposits

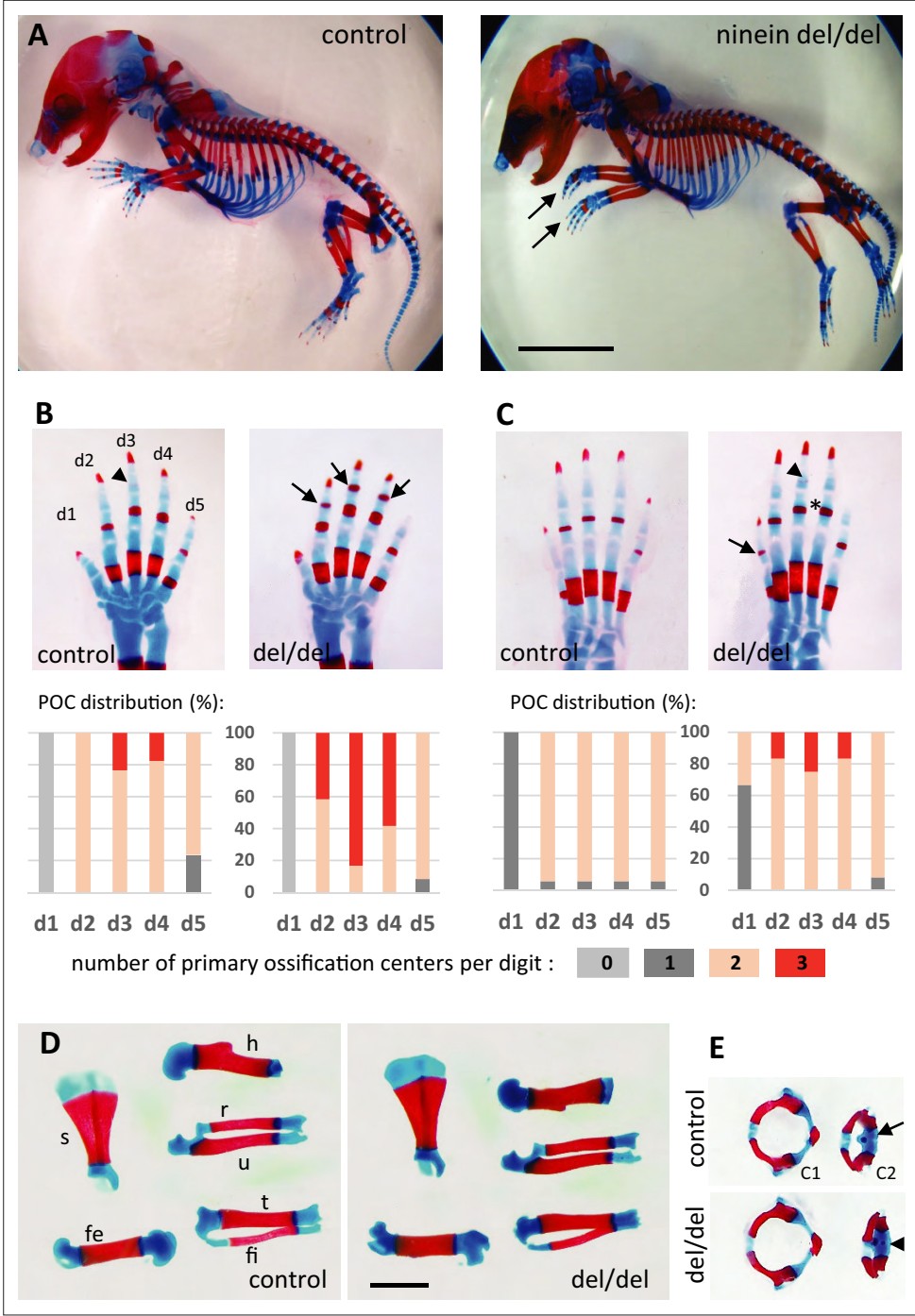

**Figure 2.** Advanced ossification in ninein-deleted mice. (**A**) Whole skeleton preparations of E18.5 embryos using Alcian blue staining for cartilage and Alizarin red staining for mineralized bone. Left: control (heterozygous), right: ninein del/del embryo. At low magnification, only curved digits from the forelimb are noticeable in ninein-deleted embryos (arrows). (**B, C**) Closer examination of the digits (d1 to d5) of both (**B**) forelimb and (**C**) hindlimb is provided for controls and ninein (del/del) embryos. Forelimb and hindlimb digits d2 to d4 show enhanced mineralization in the ninein-deleted group. Arrows in B point towards mineralization in the intermediate phalange of digits 2–4 of the forelimb, which is barely seen in digit 3 of controls (arrowhead). In the hindlimb, proximal phalanges of digits 3–5 of ninein-deleted embryos display a more intense bone staining (asterisk in C) as compared to controls. In C, the arrow points towards mineralization of digit 1 and the arrowhead indicates bone collar detection in the intermediate phalange in ninein-deleted embryos, which are absent in controls. Graphs below B and C represent the distribution of the number of primary ossification centers (POC) found for

*Figure 2 continued on next page*

*Figure 2 continued*

each digit (in phalanges, meta-carpal, and metatarsal bones), in 16 controls and 12 ninein del/del E18.5 embryos. (**D**) Dissection of long bones reveal no size differences between control and ninein-deleted embryos (fe, femur; fi, fibula; h, humerus; r, radius; s, scapula; t, tibia; u, ulna). (**E**) The second cervical vertebrae (**C2**) display one ossification center in controls (arrow) whereas an additional center of ossification (arrowhead) is present within the vertebral body of C2 in the ninein-deleted embryo. (c1, first cervical vertebrae [atlas]; c2, second cervical vertebrae [axis]). Bars, (**A**) 5 mm, (**D**) 2 mm.

on the internal face (*Figure 3B*, left), but not within central metatarsal bones yet (*Figure 3C*, left). In ninein-deficient embryos, bone collar formation was already well visible in metacarpal and metatarsal bones (*Figure 3B and C*, right). To assess long bone growth and development, we focused on the tibia metaphyseal region. We measured the height of the zones of proliferating and hypertrophied chondrocytes on longitudinal sections of embryonic tibiae as depicted in *Figure 3D*. Analysis at E16.5 and E18.5dpc revealed comparable sizes of both zones in controls and ninein del/del embryos (*Figure 3H, I*), consistent with comparable lengths of bones from forelimbs and hindlimbs at E16.5 (*Supplementary file 1*). Detection of osteoblasts, the bone-making cells, was performed by alkaline phosphatase staining and revealed no differences between both genotypes (*Figure 3E and F*). This was further analyzed on sections stained with von-Kossa and Toluidine blue stains (*Figure 3G*). Quantification of osteoblasts per bone length revealed similar numbers in embryos of both genotypes at E16.5 and E18.5 (*Figure 3J*), suggesting that differences in ossification occur independently of osteoblasts in ninein-deficient embryos.

## Premature ossification occurs at multiple sites of bone formation in ninein-deleted mice

Since maternal delivery failure was observed up to 22 days post coitum (dpc) following homozygous del/del crossing, we collected the dead pups and analyzed their skeleton to look for additional sites of enhanced bone formation at this later stage of development. We compared these pups to time-matched wild-type neonates. As shown in the supplement to *Figure 3*, the tip of the forelimb exhibited enhanced mineralization in both proximal and intermediate phalanges of digits 2–5 of ninein del/del embryos (*Figure 3—figure supplement 1A*). In the hindlimb of ninein del/del embryos, tarsal bones displayed increased mineral deposits as well as the phalanges, although to a lesser level than in forelimbs (*Figure 3—figure supplement 1B*). Bones of the rib cage did not show modified mineralization. By contrast, in the neck area, the hyoid bone always displayed primary ossification centers within the greater horns in ninein-deleted pups, which was not observed yet in controls at this age (*Figure 3—figure supplement 1C*). Among the seven cervical vertebrae, the axis contained two small mineralized areas within the anterior cartilaginous arch in controls that were significantly enlarged in ninein-deleted embryos (*Figure 3—figure supplement 1D*). Likewise, primary ossification centers in neural arches, cervical vertebrae 3–7, were enlarged in ninein-deleted embryos (*Figure 3—figure supplement 1D*). In addition, at the most caudal end of the skeleton below the lumbar 5 vertebra, caudal vertebrae had an increased ossification clearly visible within the tail of ninein-deleted embryos (*Figure 3—figure supplement 1E*). Altogether, in ninein-deleted embryos, bone mineralization took place earlier than in controls, but this occurred without interfering with endochondral bone growth.

Finally, because microcephalic phenotypes were reported in patients with compound heterozygous mutations of the ninein gene (*Dauber et al., 2012*), we analyzed littermate skull development at E16.5. Coronal head sections through the eyes revealed a more widespread mineralization in ninein-deleted embryos (*Figure 4*). Notably, frontal bones above the eyes were more intensely mineralized in ninein-deleted embryos than in controls (*Figure 4A*). Similar results were obtained in mandibular and palatal bones. Sections more posterior to the eyes revealed frontal bones with a larger trabecular structure that extends toward the interfrontal suture in ninein-deleted embryos (*Figure 4B*). Dissection of skull bones coupled with area determination showed larger frontal and parietal bones in skull of E16.5 mutant embryos (*Figure 4C and D*). The interparietal bone was slightly longer and twice larger (*Figure 4C*). Examination of the 21.5dpc mutant pups that died at delivery allowed us to look at the interfrontal suture at a later stage of development. Although the head circumference was of the same magnitude as in time-mated neonate controls, the interparietal, parietal and frontal bones were larger in mutants, whereas the supraoccipital bone was of the same size (*Figure 4E*). Most

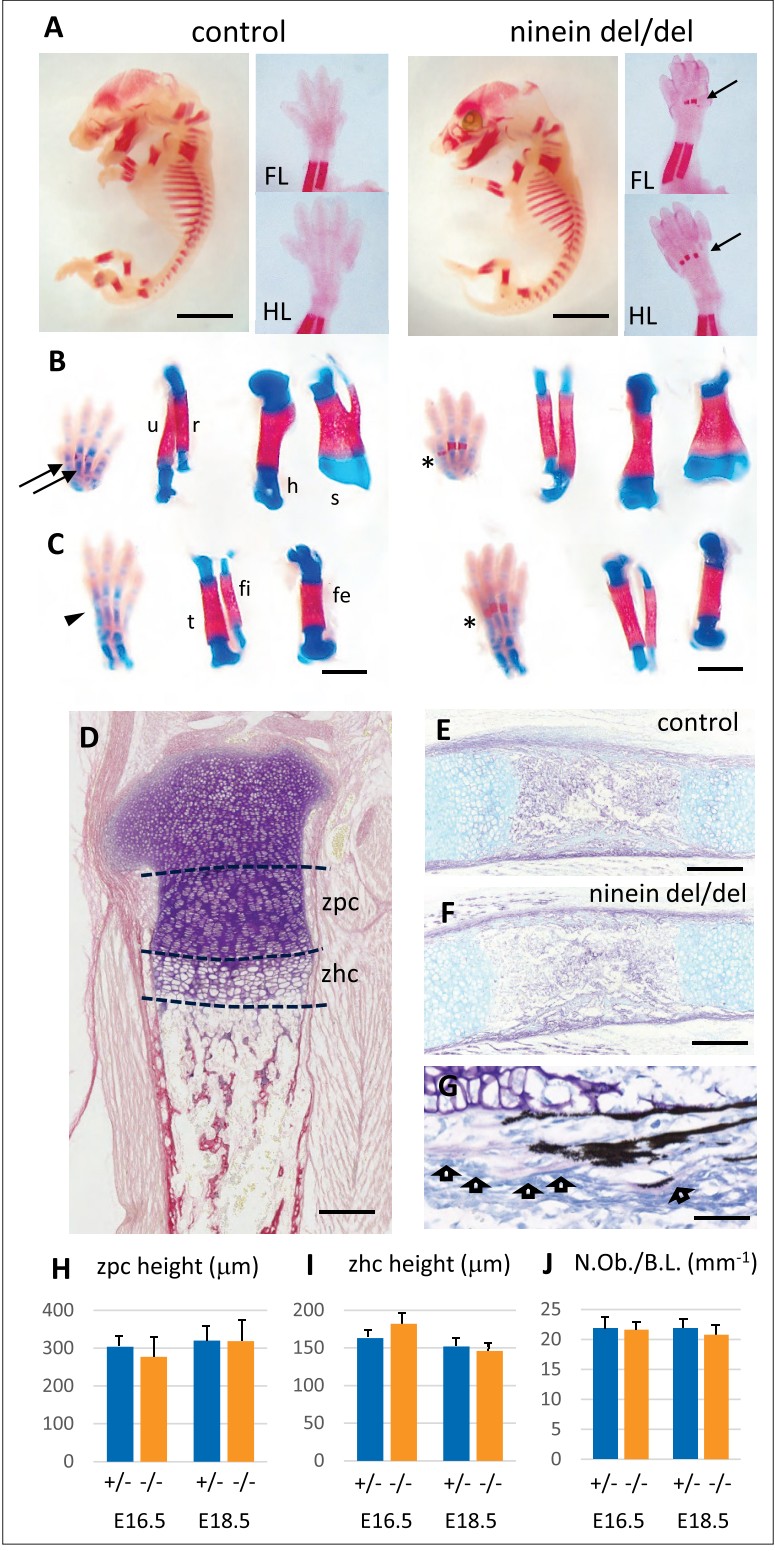

**Figure 3.** Advanced endochondral ossification in ninein-deleted mice. (**A**) Whole skeleton preparation of E16.0 embryos, stained for mineralized bone using Alizarin red. Left, control heterozygous, right: ninein del/del embryo. Despite an overall similar staining, ossification centers are visible in ninein-deleted embryos in central metacarpal and metatarsal bones (arrows, right images) whereas they are not yet mineralized in controls (FL, forelimb; HL, hindlimb). At E16.5, dual staining for cartilage and bone was performed on whole embryos, and dissected FL and HL are shown in (**B**) and (**C**), respectively. In control feet, early signs of mineralization appear in central metacarpal

*Figure 3 continued on next page*

*Figure 3 continued*

(arrows in B) and metatarsal bones (arrowhead, **C**). A stronger mineralization is evident in ninein-deleted embryos in both forelimb and hindlimb feet (asterisks in **B, C**, right panel). Dissection of long bones reveals no size differences between control and ninein-deleted embryos (fe, femur; fi, fibula; h, humerus; r, radius; s, scapula; t, tibia; u, ulna). (**D**) Pentachrome staining of a tibia epiphysis of a control embryo at E16.5. (**E, F**) Alkaline phosphatase and alcian blue staining for osteoblasts and cartilage, respectively, revealing no differences in tibiae of control and mutant embryos. (**G**) Von-Kossa and toluidine-blue-stained section of a control embryo at E18.5. Arrows indicate osteoblasts. (**H, I**) Measurements of the zone of proliferating chondrocytes (zpc) and of the zone of hypertrophied chondrocytes (zhc), as shown in (**D**). +/- indicate heterozygous control mice, -/- indicate ninein del/del mice. Eight and ten embryos were used for each genotype at E16.5 and E18.5, respectively. (**J**) Quantification of the number of osteoblasts per bone length (N.Ob./B.L.), as displayed in (**G**). Six embryos/genotype were analyzed for each time point Bars, (**A**) 3 mm, (**C**) 1 mm, (**E, F**) 250 µm and (**G**) 150 µm.

The online version of this article includes the following figure supplement(s) for figure 3:

**Figure supplement 1.** Early endochondral ossification is present at multiple sites of bone formation.

importantly, the interfrontal suture was significantly reduced close to the parietal bones (*Figure 4F*) and obliterated between nasal bones in ninein-deleted pups (*Figure 4G*). This indicates that an earlier intramembranous ossification may lead to premature closure of the skull sutures.

## Reduced numbers of osteoclasts at early stages of bone development in ninein-deleted mice

To gain insight into the origin of the premature bone mineralization in ninein del/del embryos, we examined the expression of ninein in long bones. To this end, we used the *Nin^{tm1a}* mice that expressed the *lacZ* reporter gene in the targeted *Nin* locus (*Lecland et al., 2019*). We focused on developing E16.5 embryos, and following X-Gal staining, we identified a strong signal within large multinucleated cells near the growth plate of long bones and adjacent to osteoid matrix, visualized by von-Kossa-staining (*Figure 5A and B*). Histological staining of tartrate-resistant acid phosphatase (TRAP) was used as a marker of osteoclasts on serial sections, to confirm that the multinucleated cells were indeed osteoclasts (*Figure 5C*). We then combined TRAP analysis with von-Kossa-staining of mineralized deposits in developing tibia and at E16.5 and E18.5. These analyses were limited to the upper metaphyseal border within 400 µm below the zone of hypertrophied chondrocytes at E16.5 and within 600 µm for E18 tibiae (*Figure 5D and E*). At E16.5, a significant increase of the mineralized portion of the tibia was detected in ninein del/del embryos (*Figure 5F*). This was accompanied by a decrease in osteoclast cell numbers within the same area (*Figure 5G*). On E18.5 tibiae, increased mineralization was no longer detected although osteoclast numbers were still significantly reduced (*Figure 5G*). To further analyze tibia morphology, we used X-ray microtomography to investigate bone structure in ninein-deleted E18.5 embryos and control siblings (*Figure 5H–K*). Scanning through the cortical and trabecular bones below the hypertrophied chondrocytes revealed no difference between control and mutants (*Figure 5H, I*), in agreement with histology data. In a subchondral core of the tibia of 200 µm diameter and 700 µm length, the volume occupied by mineralized bone was of the same magnitude in control and ninein-deleted embryos ($0.00210\pm0.00026$ mm$^3$ vs $0.00203\pm0.00006$ mm$^3$, respectively), corresponding to a bone volume fraction (BV/TV) of $9.8\pm1.3$ vs $9.4 \pm 0.4$%, respectively (3D-reconstructed cores in *Figure 5H, I*). Meanwhile, analysis performed in the middle of the tibiae in controls revealed that the volume of mineralized bone had reduced by 2-fold ($0.00107\pm0.00025$ mm$^3$), consistent with the onset of bone marrow cavity formation, following blood vessel invasion in this central part of the bone (*Figure 5J*). By contrast, in ninein-deleted embryos the volume of mineralized bone in this central area was significantly higher as compared to controls ($0.00193\pm0.00012$ mm$^3$, $p<0.05$; *Figure 5J and K*). The BV/TV ratio was $4.5 \pm 1.0$% in controls vs $7.8 \pm 0.7$% in tibiae from ninein-deleted embryos, $p<0.05$. Altogether, our data suggest that the genesis of osteoclasts was impaired in the absence of ninein, resulting in structural changes in the developing trabecular bone of the tibia.

## Osteoclast progenitors display cell fusion defects in the absence of ninein

To analyze the mechanisms that lead to the decrease of osteoclast numbers in ninein-deleted embryos, we focused on early steps of cell fusion that generate multinucleated, functional osteoclasts. First,

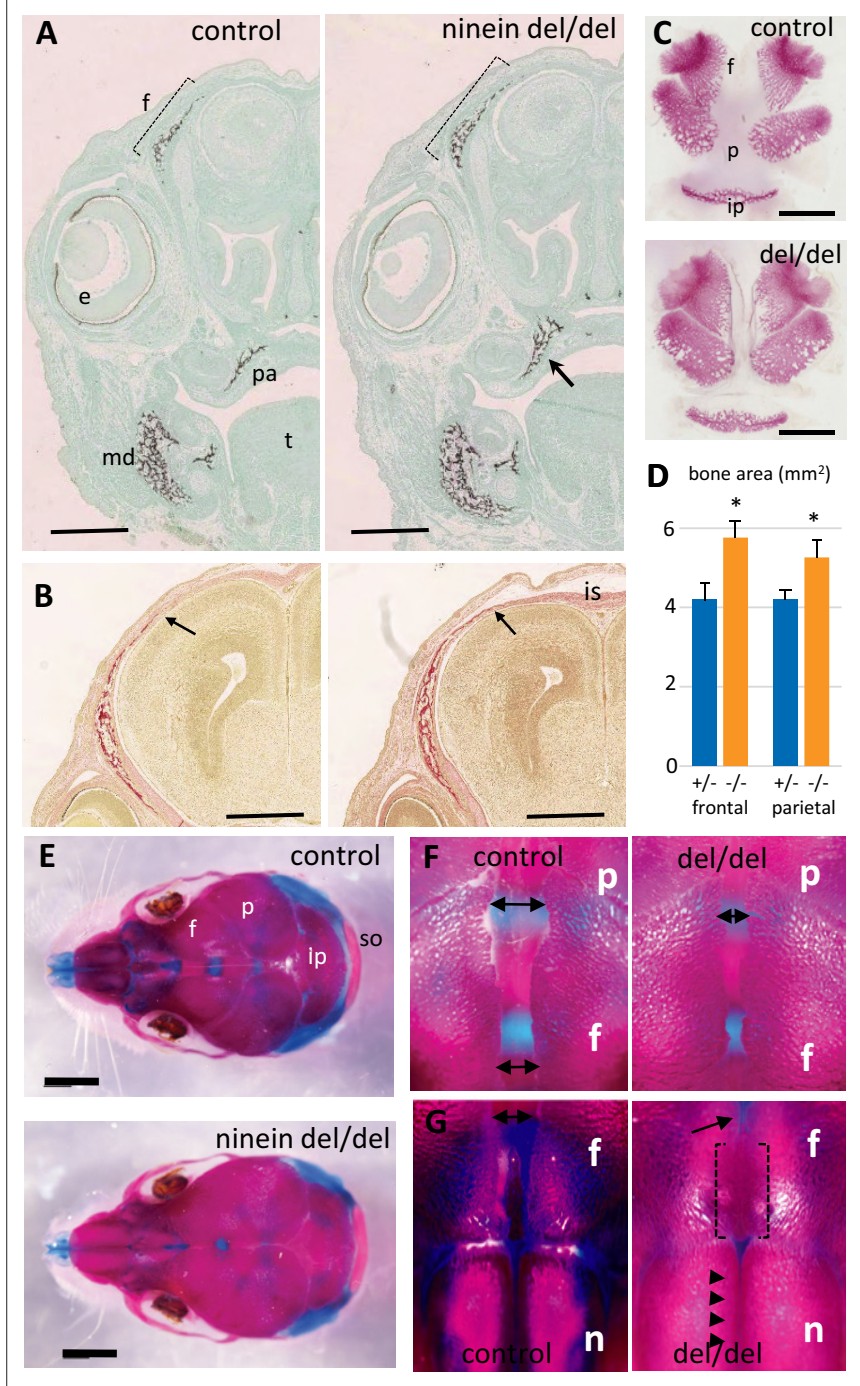

**Figure 4.** Early intramembranous ossification in ninein-deleted mice leads to premature suture closure. (**A, B**) Coronal sections through E16.5 heads of control and ninein-deleted embryos (e, eye; f, frontal; is, interfrontal suture; md, mandible; pa, palatal; t, tongue). (**A**) Von Kossa staining revealed enhanced mineralization in skull lacking ninein, particularly visible for the frontal (dotted line) and palatal bones (arrow, right). (**B**) Alizarin red S staining highlighted an enhanced trabecular frontal bone towards the interfrontal suture (arrow, right). (**C**) Dissected frontal and parietal bones from control and ninein-deleted embryos at E16.5 (ip, intraparietal; p, parietal). (**D**) Measurements of (**C**), indicating increased bone areas in ninein-deleted skulls. +/- represent heterozygous controls, -/- represent ninein del/del embryos. *, p<0.05, as compared to controls (n=4 in each group). (**E**) E21.5dpc skulls stained with Alcian blue and Alizarin red (n, nasal; so, supraoccipital). Top: control, bottom: ninein del/del. (**F**) High magnification of skulls, close to the parietal bones: a clear reduction in the space between frontal bones is observed in ninein-deleted embryos (compare double arrows left and right). (**G**) More

*Figure 4 continued on next page*

*Figure 4 continued*

anterior position of the skull. The interfrontal suture is closed in ninein del/del skulls (open brackets, right), and almost closed between nasal bones (arrowheads, right) as compared to controls. Bars, (**A, B**) 500 μm, (**C**) 2 mm, (**E**) 5 mm.

---

we followed osteoclastogenesis early in development, at E14.5, which depends on erythro-myeloid progenitors. On longitudinal sections of whole embryos, von-Kossa-staining revealed the absence of mineralization at this stage of development, both in controls and ninein del/del mice (***Figure 6—figure supplement 1A, B***). On the same sections, TRAP enzymatic reaction indicated the presence of osteoclast progenitors in the fetal liver, an active site of erythro-myeloid hematopoiesis in all E14.5 embryos. Two additional foci of TRAP-positive cells were observed in an area corresponding to the mandibular and maxillary bone formation, the first bones to mineralize (***Figure 6—figure supplement 1A, B***). We therefore focused our analysis on the initial steps of intramembranous ossification in the developing mandible. Numerous TRAP-positive cells were detected close to Meckel's cartilage, in both control and ninein-deleted embryos (***Figure 6—figure supplement 1C, D***). Because the first step of bone formation involves the production of a collagenous matrix by condensed mesenchymal cells, we used Sirius Red dye to verify that collagen deposition was effective in the mandible area, where osteoclast progenitors were detected (***Figure 6—figure supplement 1E, F***). Similarly, the presence of bone-forming osteoblasts was confirmed using alkaline phosphatase enzymatic reaction (***Figure 6—figure supplement 1G, H***). All three staining protocols yielded comparable results in control and ninein del/del siblings. Since the expression of tartrate-resistant acid phosphatase was prominent in the mandible area, we counted TRAP-positive cells and their number of nuclei, to assess early fusion events in osteoclast formation in E14.5 littermates. Images of TRAP-positive cells with one, two or three nuclei are presented in ***Figure 6A***. Morphological changes in TRAP-positive cells were observed, from small round mononucleated cells, to elongated binucleated cells and large cells with many nuclei on the surface of the osteoid matrix, as recently published (***Nakamura et al., 2021***). Quantitative analysis of approximately 1300 cells/genotype showed that mononucleated osteoclast progenitors represented the vast majority of TRAP-positive cells (***Figure 6A***). They were significantly more abundant in ninein-deleted embryos, as compared to control littermates. Similar percentages of binucleated osteoclasts were counted in control and ninein-deleted E14.5 embryos. However, multi-nucleated osteoclasts were more than threefold reduced in ninein-deleted embryos as compared to controls (***Figure 6A***), indicating that the initial fusion events among osteoclast progenitors were impaired at E14.5.

Next, we investigated whether any ninein-dependent fusion defects remained after birth, when osteoclasts are formed from hematopoietic stem cell precursors. Bone marrow cells from control and ninein-deleted newborn mice were seeded on glass coverslips, of which osteoclast precursors sponta-neously adhered to the glass surface, whereas non-adherent cells (more than 98% of the cells initially seeded) could be rinsed off after one hour. TRAP-positive cells were classified in four categories: (i) single mononucleate cells, (ii) aggregates of cells with sporadic intercellular contacts with generally a TRAP-positive cell in the center, (iii) large clusters of cells with larger contact areas, reminiscent of cells engaged in the fusion process, and (iv) multinucleated osteoclasts (***Figure 6B***). These four categories resembled the classical steps of osteoclastogenesis, as previously described (***Boyle et al., 2003***). Quantification of each category indicated that multinucleated osteoclasts were significantly less abundant in samples from ninein-deleted neonates, whereas single TRAP-positive cells were more numerous, in comparison to controls (***Figure 6B***). The percentages of aggregates with unfused or fused cells were similar in both groups.

To further explore the possibility of ninein-dependent fusion defects among osteoclast precursors, we used cultures of adult bone marrow from femur, as described (***Vérollet et al., 2013***). Fusion of precursor cells into syncytia occurred within 3 days in the presence of RANK-L. From day 3 onwards, control osteoclasts reached diameters of several hundred microns, and the increase in cell surface correlated in a linear fashion with the number of syncytial nuclei (***Figure 7A and B***). The largest syncytia contained more than 100 nuclei, consistent with other reports on mouse osteoclasts in culture (***Tiedemann et al., 2017***). A comparable correlation was seen in osteoclasts from ninein-deleted bone marrow, but larger syncytia were significantly less abundant in these mutants (***Figure 7B***, ***Figure 7—figure supplement 1A, B***). To investigate the fusion competence of control and ninein-deficient

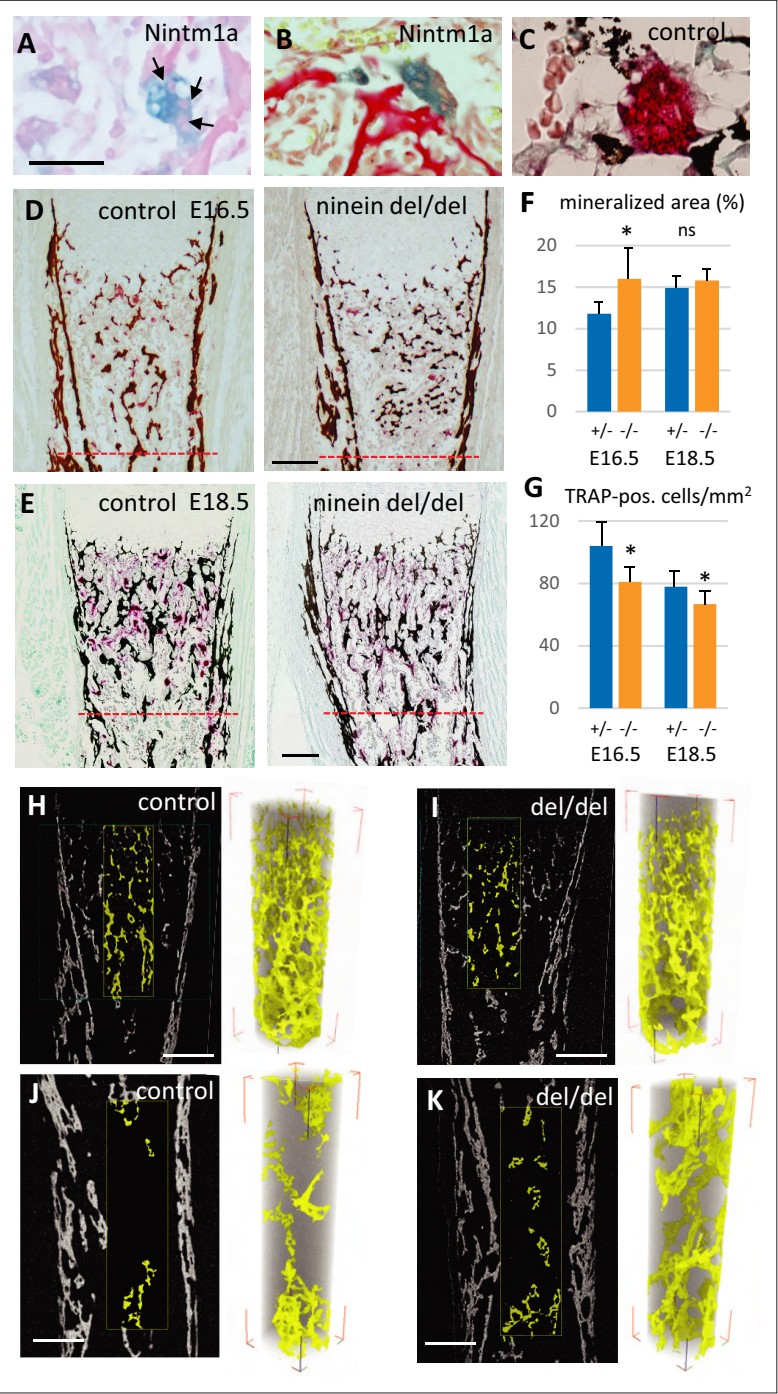

**Figure 5.** Reduced osteoclast density affects long bone internal structure at E18.5. (**A**) Histology of E16.5 heterozygous Nintm1a mice revealed high expression of β-galactosidase (blue) in multinucleated cells (arrows). (**B**) These β-galactosidase-positive syncytia line close to alizarin red stained material. (**C**) The syncytia are positive for TRAP (red), an osteoclast-specific enzymatic reaction. Black color represents von Kossa-staining. (**D**) Tibiae from E16.5 and (**E**) E18.5, left: control, right: ninein del/del. Images depict von Kossa staining (black), followed by TRAP enzymatic reaction (red). (**F**) Analysis of the mineralized bone area was performed within 400 μm and 600 μm below the zone of hypertrophied chondrocytes, limits represented by the red dotted lines in (**D, E**). +/- represent heterozygous controls, -/- represent ninein del/del embryos. (**G**) In the same area, TRAP-positive cells were quantified. A significant reduction in the number of osteoclasts per square millimeter was observed in ninein del/del embryos. An increase in the mineralized bone was detected in ninein del/del samples only at E16.5. *, p<0.05 as compared to controls. ns, statistically not significant. Data on E16.5 tibiae (**F**) were from 7 controls and 8

*Figure 5 continued on next page*

*Figure 5 continued*

ninein-deleted embryos. Data on E18.5 tibiae (**G**) were from 9 controls and 9 ninein-deleted embryos. (**H, J**) X-ray microtomography was performed on E18.5 control and (**I, K**) ninein-deleted whole microdissected tibiae. The upper parts of the mineralized tibiae area (yellow colored) were of the same density in both groups. Within the middle part of the tibiae, the central core of mineralization represented in yellow was denser in ninein del/del tibia as compared to control. Bars, (**D**) 100 μm, (**E**) 150 μm, (**H–K**) 200 μm.

osteoclast cultures, we determined the fusion indices, by quantifying the number of syncytial nuclei relative to the total number of nuclei in the culture. We found that the fusion index of osteoclasts from ninein-knockout mice was 1.8 times lower than in cultures from control mice (9% vs 16%, *Figure 7C*). Likewise, multinucleated osteoclasts in ninein-deficient cultures covered an area that was 1.9 times smaller than in control cultures, with the total density of nuclei (combined from fused and unfused cells) being comparable for the two genotypes (*Figure 7C*). To verify whether reduced fusion in ninein-deficient cells was resulting from differences in proliferation and cell cycle progression, we established cell cycle profiles (*Figure 7D*), and we quantified the expression of the proliferation marker Ki-67 (*Figure 7—figure supplement 1C*) one day before the onset of fusion. No significant differences between control and mutant cultures were detected. To exclude the effect of cell death on osteoclastogenesis in ninein-deficient cultures, we performed co-staining with Annexin V and propidium iodide, and we controlled for the presence of cleaved poly-(ADP-ribose)-polymerase (PARP) and cleaved caspase 3 (*Figure 7—figure supplement 1D, E*). Again, no differences were seen between the different genotypes. We could further exclude any significant differences in cell attachment of the osteoclast precursor cells, since numbers of adherent cells and expression profiles of beta-integrins were comparable prior to osteoclast fusion (*Figure 7—figure supplement 1F, G*). In summary, our data indicate that ninein-deficient osteoclast precursors exhibit specific defects in cell fusion during osteoclastogenesis. Next, we quantified the abundance of large syncytia (from 5000 to 50,000 μm$^2$) on days 3, 4, and 5 in culture, following TRAP staining. On average, these large syncytia represented less than 5% of osteoclasts of both genotypes (*Figure 7—figure supplement 1A*), but they were twice as abundant in control cultures on days 3 and 4, as compared to ninein-deleted samples (*Figure 7—figure supplement 1B*). However, on day 5, the abundance of large syncytia lacking ninein climbed to more than 90% of control levels, suggesting that fusion deficiencies are a transient phenomenon in in vitro-induced adult osteoclasts. On later days of culture, fusion efficiency diminished and osteoclasts, known to be short-lived in vitro (*Akchurin et al., 2008*), started to die. Importantly, when osteoclast function was assessed in bone resorption assays, we found that the resorption area of osteoclasts lacking ninein was approximately three times smaller than that of controls (*Figure 7E*).

## Reduced centrosome cohesion and clustering in osteoclasts lacking ninein

Because ninein is primarily known as a microtubule-organizing protein at the centrosome, we investigated whether osteoclast precursors from newborn mice showed any centrosomal abnormalities. As expected, precursor cells isolated from bone marrow of ninein-deleted neonates were negative for ninein (*Figure 8A*). Staining with the centriolar marker centrin revealed that mother and daughter centrioles were separated to varying degrees in the absence of ninein (mean distance 1.1 μm±0.2, n=11 different mice), whereas in control cells they remained more closely associated (0.6 μm±0.1, n = 7 different mice; *Figure 8A*). Microtubule organization at the centrosome appeared less focused in cells without ninein, consistent with previously published results (*Dammermann and Merdes, 2002*; *Figure 8B*). Because ninein interacts with the dynein/dynactin complex, and because dynein/dynactin itself possesses a microtubule-focusing activity, we tested for the presence of the dynactin subunit p150 in ninein del/del cells (*Quintyne et al., 1999*; *Casenghi et al., 2005*). Immunofluorescence revealed a reduction of 48% of the p150 signal at centrioles in these cells (n=52; *Figure 8C*). The defects in precursor cells from bone marrow of adult ninein-deleted mice resembled those seen in cells from newborn mice, and loss of microtubule focusing was even more pronounced (*Figure 8D and E*). Upon fusion into multinucleated osteoclasts, significant differences in centrosome positioning became apparent in ninein-deleted cells. Each fusing precursor cell contributes one centrosome, that is two centrioles, to the syncytium. Whereas the numerous centrioles in control osteoclasts were all regrouped in one or very few clusters, the individual centrioles in osteoclasts lacking ninein were

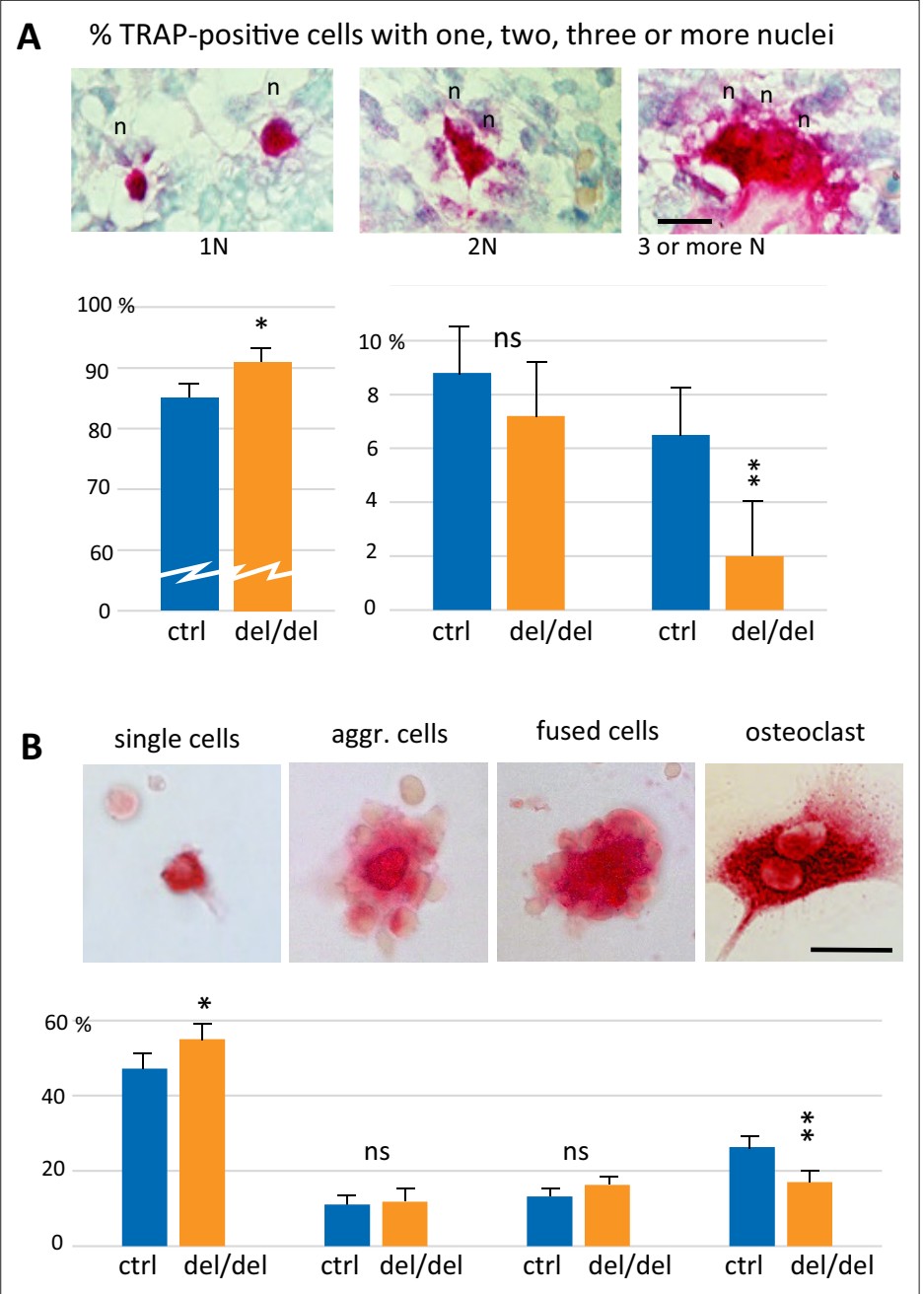

**Figure 6.** Reduced cell fusion of osteoclast precursors during osteoclastogenesis. (**A**) Focus on mandible development from sections of E14.5 control and ninein-deleted embryos. Examination of TRAP-positive cells within the mandible area, revealing the presence of mononucleated cells (1 N), binucleated cells (2 N), and cells with three or more nuclei (3 N; n, nucleus). Graphs indicate the percentage of TRAP-positive cells with one, two or ≥three nuclei. 1300 cells were analyzed for each genotype (control [ctrl], ninein del/del). * and **, p<0.05 and p<0.01, respectively, by comparison of ninein del/del to controls (n=8 controls and 7 ninein-deleted embryos; ns, statistically not significant). (**B**) Neonatal bone marrow cells cultured for one hour and stained for TRAP. TRAP-positive cells are classified in four groups: single cells, aggregated cells (aggr. cells), clusters of fused cells, and osteoclasts. The percentage of cells in each category is presented below each image. In samples from ninein del/del bone marrow (dark blue bars), an increased percentage of mononucleated TRAP-positive cells, and a reduced percentage of mature osteoclasts are detected (* and **, p<0.05 and p<0.01, respectively, as compared to controls (ctrl, light blue bars). Data are from 16 controls and 15 ninein-deleted neonates; ns, statistically not significant). Bars, (**A, B**) 25 µm.

*Figure 6 continued on next page*

*Figure 6 continued*

The online version of this article includes the following figure supplement(s) for figure 6:

**Figure supplement 1.** Early osteoclastogenesis from erythro-myeloid progenitors.

scattered or localized close to the surfaces of the syncytial nuclei (*Figure 8F and G*). Large clusters of centrioles in control osteoclasts represented focal points of microtubule organization and dynactin accumulation, whereas no comparable focusing was visible in ninein del/del osteoclasts (*Figure 8F and G*). Quantification of centriole clustering (*Figure 8H*) was performed as described by *Phan et al., 2004*, revealing that 93% of control osteoclasts displayed at least one cluster, whereas only 11% of ninein del/del osteoclasts showed any clustering. Moreover, multiple clusters were visible in medium and large control osteoclasts, but completely absent from any ninein del/del osteoclasts. The occasional appearance of a single cluster in mutant osteoclasts of any size may be due to random proximity of centrioles, rather than active clustering.

## Discussion

Using ninein-deficient mice, we show that the absence of the centrosomal protein ninein induces moderate skeletal abnormalities, accompanied by a permanent reduction of osteoclastogenesis from early embryogenesis to adulthood. We found that osteoclasts are very sensitive to the loss of ninein, thus revealing an unexpected role of ninein in bone formation. Although the bone abnormalities we describe here are apparently minor, we show that they have a pronounced effect on the closure of the skull suture, which may ultimately lead to craniosynostosis. Multiple craniosynostosis-related syndromes have been described in humans, most of them secondary to impaired signaling by FGF or BMP (*Zhao et al., 2023*; *Ueharu and Mishina, 2023*). Excess osteogenic differentiation of suture mesenchymal cells or defects of stem cells in sutures is usually thought to be the leading cause of craniosynostosis (*Ueharu and Mishina, 2023*). Interestingly, craniosynostosis has been reported for a group of patients with microcephalic Seckel syndrome (*Parent et al., 1996*). Here we propose that osteoclast defects might be a novel element contributing to microcephaly in humans upon ninein mutation, in addition to proliferative defects that impact on the overall growth of the body and the brain (*Dauber et al., 2012*).

### Lack of multinucleate osteoclasts as a cause for ossification defects

Recent publications on the developmental origin of osteoclasts have highlighted the various cell lineages involved in osteoclastogenesis (*Jacome-Galarza et al., 2019*; *Yahara et al., 2020*; *Yahara et al., 2021*). Embryonic and neonatal osteoclasts can derive from embryonic precursors such as yolk-sac-derived macrophages, fetal liver myeloid progenitors, and/or hematopoietic stem cells precursors. The initial fusion of osteoclasts that we observed at E14.5 likely involved the population derived from erythroid-myeloid progenitors, since these are the first to be generated at mid-gestation. We show that for intramembranous ossification of the mandible at this early stage, numerous mononucleated osteoclast progenitors and osteoblasts are already present along collagen fibers prior to initiation of mineralization, consistent with previous studies (*Nakamura et al., 2021*). The reduced percentage of multinucleated osteoclasts in the ninein-deleted embryos likely disturbs the balance between bone formation and degradation, as mononucleated osteoclasts are thought to have a lower capacity of bone resorption. This was emphasized in an earlier report on a microphthalmic osteopetrotic mouse in which mononucleated osteoclasts result from fusion defects in very young embryos (*Thesingh and Scherft, 1985*). These osteoclasts function inefficiently in bone resorption at prenatal stages. One hallmark of osteopetrosis is the failure of forming a normal bone marrow cavity, which usually involves osteoclasts formed from early and late erythroid-myeloid progenitors (*Yahara et al., 2021*). These progenitors give rise to long-lasting osteoclast precursors that contribute to postnatal bone remodeling through fusion with hematopoietic-derived osteoclasts. In our ninein del/del embryos, we observe delayed bone cavity formation in prenatal tibiae, although we do not detect osteopetrosis. Notably, the microphthalmic osteopetrotic mouse is able to produce multinucleated osteoclasts in the bone marrow ten days after birth, suggesting that a more mature bone marrow may compensate for earlier defects (*Thesingh and Scherft, 1985*). Similarly, the fusion defects in osteoclast cultures from

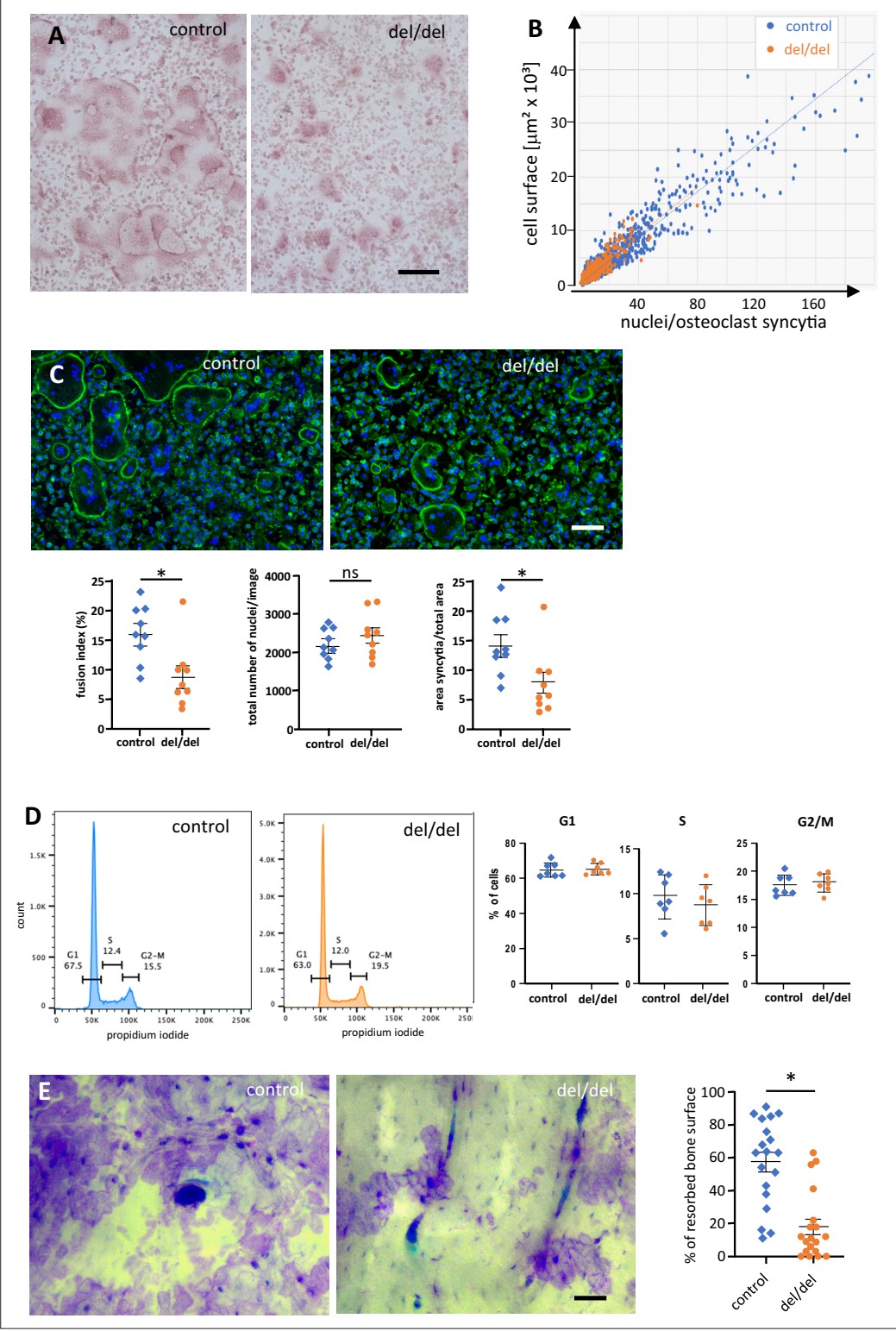

**Figure 7.** Reduced cell fusion of osteoclast precursors from cultured bone marrow cells. (**A**) TRAP staining of osteoclasts from adult mouse bone marrow at day 3 of culture, revealing size differences between controls and ninein del/del. (**B**) Fusion of precursors cells from bone marrow into osteoclasts, as seen in (**A**). Correlation of the cell surface with the number of nuclei/syncytium. Large syncytia are significantly less abundant in ninein del/del samples (orange dots). (**C**) Fusion assay of osteoclast precursors in culture. Representative images of cultures from control and ninein del/del mice, stained with phalloidin-Alexa 488 (F-actin, green) and DAPI (DNA, blue).

*Figure 7 continued on next page*

*Figure 7 continued*

Corresponding graphs depict the fusion index (number of nuclei in syncytia relative to the total number of nuclei), the total number of nuclei/image, and the area covered by syncytial osteoclasts/total area. For each graph, 3 fields of view (1,700,000 $\mu m^2$) of each culture from 3 mice were quantified for each genotype. The data were tested for normality using a Shapiro-Wilk test, and a Mann-Whitney test was applied to the quantification of the fusion index and of the syncytial area. Mean ± SEM are indicated. *, p<0.05, ns: not significant. (**D**) Flow cytometry analysis of osteoclast precursors from bone marrow, one day prior to fusion. Right: quantification of propidium iodide staining in G1, S, and G2/M phases was performed from samples from 3 mice/genotype (to limit pipetting errors, two to three samples were analyzed from each mouse). (**E**) In vitro-bone-resorption assays. Left: resorption pits from osteoclast cultures from control and ninein del/del mice. Right: quantification of the resorption areas, from image analysis of 6–7 fields of view of each culture from 3 mice/genotype. *, p<0.05. Bars, (**A, C**) 100 $\mu m$, (**E**) 50 $\mu m$.

The online version of this article includes the following source data and figure supplement(s) for figure 7:

**Figure supplement 1.** Time course of osteoclast fusion, and rates of proliferation, viability, and cell adhesion of precursor cells.

**Figure supplement 1—source data 1.** Immunoblots of osteoblast precursors from control and ninein del/del mice, one day before fusion, probed with antibodies against poly-(ADP-ribose)-polymerase (PARP) and cleaved caspase 3, and probed with antibody against Glyceraldehyde-3-phosphate dehydrogenase (GAPDH) as a loading control.

ninein del/del mice appear to be transient, as longer times of cell culture allowed for compensation of these defects.

## Cellular defects underlying reduced osteoclastogenesis

Ninein-dependent defects may be responsible for reduced cell proliferation or cell death that may, at least in part, restrain growth at very early embryonic stages in ninein-KO mice. Abnormalities of the mitotic spindles may be at the origin of these defects, since ninein has been reported to localize to the spindle poles and to prevent the formation of multipolar spindles (*Logarinho et al., 2012*). Consistently, loss of a ninein homologue in *Drosophila* increases the frequency of mitotic defects, even though the gene is dispensable for the viability of the fly (*Kowanda et al., 2016*; *Zheng et al., 2016*). Besides playing a role in mitotic fidelity, ninein has been shown to contribute to spindle orientation and may therefore help maintaining the pool of proliferative progenitor cells in asymmetric divisions (*Lecland et al., 2019*). However, we demonstrate that specifically in osteoclastogenesis, the absence of ninein affects directly the fusion process of mononucleated osteoclast precursors into syncytial osteoclasts, whereas cell proliferation, attachment, and viability of precursors in culture remain unaffected by the ninein-knockout. An efficient fusion process is thought to require the proximity of cells, which depends on directed cell migration. Directed migration is supported by a centrosomal microtubule-organizing center that grows microtubules towards the cell's leading edge, and to the uropod in cells undergoing amoeboid movements (*Luxton and Gundersen, 2011*; *Kopf and Kiermaier, 2021*). Loss of the centrosomal organizing center interferes with directional migration, whereas reinforcement by amplification and clustering of centrioles enhances directional migration (*Wakida et al., 2010*; *Weier et al., 2022*). The fusion of precursor cells into osteoclasts is finally mediated by intercellular contacts. Fusions are thought to require local polarization of the cells and the formation of microtubule-dependent cell extensions (*Straube and Merdes, 2007*; *Dufrançais et al., 2021*). We hypothesize that both directional cell migration and fusion require a strongly focused microtubule array, organized from a single centrosomal center. In osteoclast precursors from ninein-KO mice, we observe loss of microtubule attachment at the centrosome, and splitting of the organizing center by separation of the two centrioles. These findings are consistent with similar observations in other cell types (*Dammermann and Merdes, 2002*; *Mazo et al., 2016*; *Theile et al., 2023*). We believe that these defects interfere with the directionality of radial microtubule growth towards the cell periphery and thereby interfere with the directional transport of membrane vesicles, during cell migration and during the formation of plasma membrane extensions in fusing osteoclasts. In a somewhat analogous manner, cell polarity in neurons and the initial formation of the axonal cell extension depend on centrosomal microtubules (*de Anda et al., 2005*).

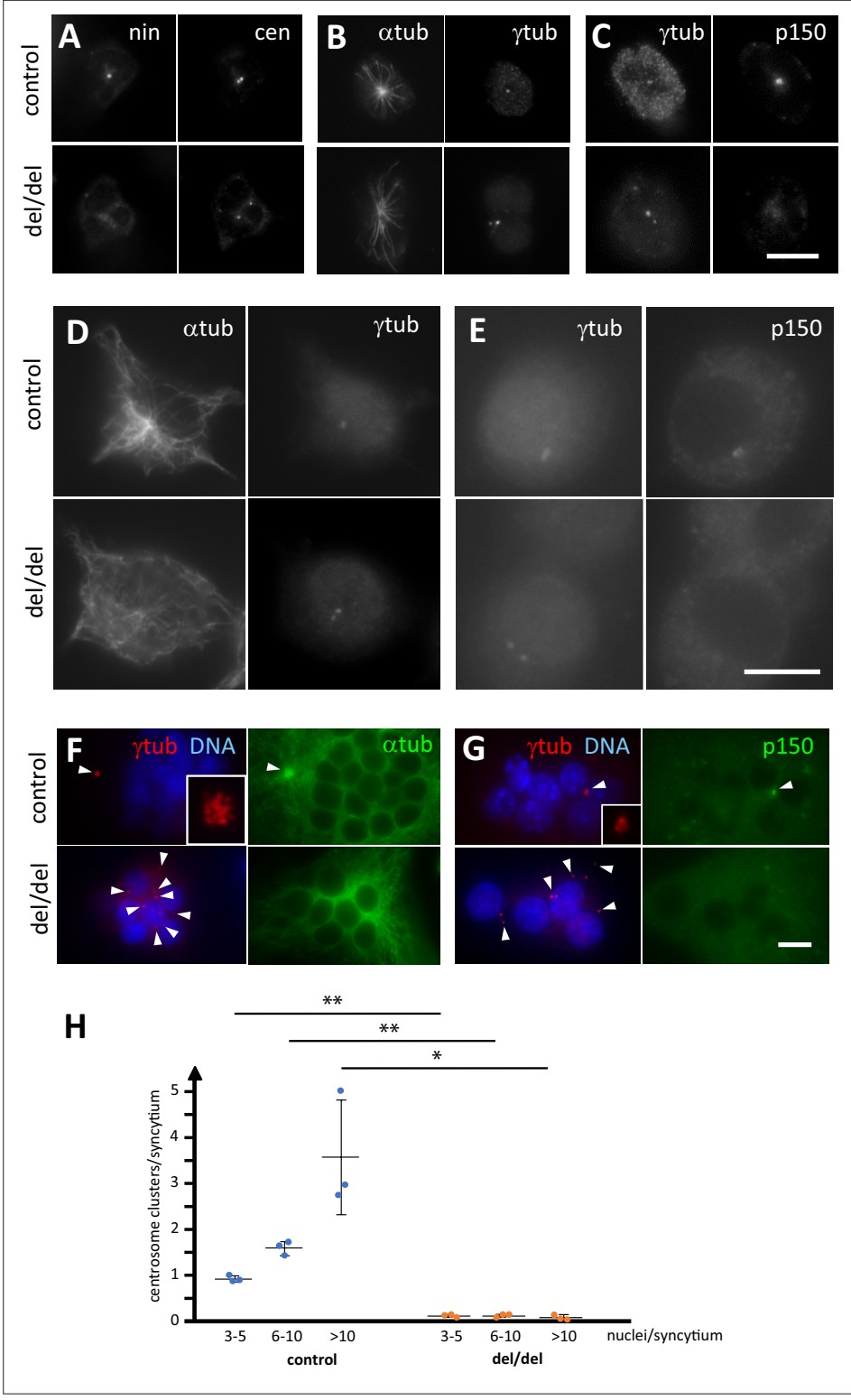

**Figure 8.** Abnormal centriole clustering in osteoclast cultures from ninein-deleted animals. (**A**) Double immunofluorescence of neonatal osteoclast progenitors for ninein (nin) and centrin (cen), (**B**) α and γ-tubulin, and (**C**) γ-tubulin and dynactin-subunit p150. Controls (upper images) and ninein-deleted cells (lower images, del/del) are shown. (**D, E**) Osteoclast precursors obtained from adult bone marrow cell culture, after 2 days in the presence of M-CSF and RANKL, labelled for (**D**) α and γ-tubulin, or (**E**) γ-tubulin and dynactin p150. Controls (upper images) and ninein-deleted cells (lower images, del/del) are shown. (**F**) In vitro-differentiated osteoclasts obtained from the

*Figure 8 continued on next page*

*Figure 8 continued*

culture of adult bone marrow cells after 3 days, labelled for α and γ-tubulin, or (**G**) γ-tubulin and dynactin p150, and nuclei stained with DAPI (DNA). The lower panels are from ninein del/del adult bone marrow and display lack of clustering of centrioles (arrowheads), as compared to controls that show clustering of the majority of centrioles in a single focus. Microtubules are focused in the area of centriole clustering (F, single arrowhead), and dynactin p150 is enriched in the cluster (**G**) in controls only (single arrowhead). Insets show enlarged views of the centriole clusters. (**H**) The average number of centriole clusters in differentiated osteoclast cultures was determined. Clusters were defined as having 3 or more signals of γ-tubulin within 1 µm of each other. Three categories of osteoclasts were distinguished: small (3–5 nuclei), medium (6–10 nuclei), and large ones (>10 nuclei). The average number of clusters/syncytium is shown for each mouse by blue or orange circles, from cultures of n=3 mice/genotype (approximately 300 osteoclasts/coverslip were analyzed). *, p<0.01; **, p<0.05. Bars, (**A–F**) 10 µm.

## Ninein-dependent protein interactions during osteoclastogenesis

At the molecular level, ninein-dependent microtubule anchorage and centrosome cohesion may depend, at least in part, on the interaction with the dynein/dynactin complex. An amino-terminal domain of ninein is known to bind to dynein/dynactin and to play a role in dynein activation (*Casenghi et al., 2005*; *Redwine et al., 2017*). Depletions of the dynactin subunit p150 or of the dynein heavy chain cause splitting of the centrioles (*Kodani et al., 2013*; *Malik et al., 2016*). Moreover, depletions of the dynein regulator proteins Lis1 and Nde1 interfere with the motility and fusion of osteoclast precursors, and Lis1-depletion also leads to altered microtubule organization in osteoclasts (*Ye et al., 2011*; *Das et al., 2021*). A dynein/dynactin-dependent mechanism may equally contribute to the clustering of the numerous centrioles observed in multinucleated osteoclasts (*Phan et al., 2004*). In addition to perinuclear microtubule-organizing centers, these centriole clusters provide large focal points for microtubule organization and may thereby support cell extensions, for fusion with additional precursors into even larger syncytia (*Mulari et al., 2003*; *Phan et al., 2004*). Furthermore, clustered centrioles appear to be necessary for osteoclast polarity and secretory processes (*Lee et al., 2023*). As we demonstrate here, the quantity of dynactin at the centrosome is largely reduced in the absence of ninein, and clustering of centrioles fails to take place. As in mononucleated precursor cells, a self-centering force on the centrioles, dependent on ninein-bound dynein/dynactin may be required for this clustering process. In addition, ninein may play a direct role as a linker component between centrioles, as suggested from recent experiments in various human cell lines (*Theile et al., 2023*). The absence of ninein may therefore explain the clustering defect we observed in mature osteoclasts. Interestingly, homozygous missense mutations in the ninein gene of patients with spondyloepimetaphyseal dysplasia were shown to alter the carboxy-terminal domain of ninein (N2082D), in an area that mediates the interaction with the centriolar linker protein cNap1 (*Grosch et al., 2013*; *Zhang et al., 2016*). Since cNap1 is directly involved in maintaining centrosome cohesion (*Mayor et al., 2000*), it is conceivable that centriole-splitting contributes to ossification defects. Furthermore, a truncation mutation in cNap1 in cows is at the origin of centriole splitting and cell migration defects, as well as Seckel-like syndromes (*Floriot et al., 2015*). Clustering of centrioles may also contribute to the functionality of placental cells and thereby insure early embryonic growth, since specialized cells of the embryonic trophoectoderm, trophoblast giant cells, are polyploid and display centrosome amplification (*Klisch et al., 2017*; *Buss et al., 2022*), with clustered centrioles identified in binucleate bovine trophoblast cells (*Klisch et al., 2017*). Thus, early embryonic growth defects in our ninein-deficient mouse embryos may be explained in part by impaired centriole clustering, in addition to potential abnormalities in proliferation and cell death.

Besides contributing to centrosome cohesion, ninein may regulate the number of microtubules attached at the centrosome (*Dammermann and Merdes, 2002*), either by holding onto microtubule minus-ends via dynein/dynactin, or via the ninein-interactor Cep170 that has direct microtubule-binding affinity (*Quintyne et al., 1999*; *Welburn and Cheeseman, 2012*). Ninein may thus anchor microtubules to the subdistal appendages, as well as to the proximal regions of the centriolar surface (*Mogensen et al., 2000*). However, in contrast to earlier proposals (*Delgehyr et al., 2005*), ninein may not affect the nucleation of centrosomal microtubules, as we see no significant reduction of gamma-tubulin at the centrosomes of ninein-depleted cells. Independent of its centrosomal function, ninein may play a role in the recruitment of F-actin fibers to cortical sites by interacting with dynein/dynactin,

as highlighted in a recent study on macrophages (*Omer et al., 2024*), raising the question whether ninein-knockout may cause subtle alterations in the actin cytoskeleton in osteoclast precursors.

## Conclusion

Altogether, the ninein-dependent defects at the cellular level are subtle, since osteoclast fusion still takes place, albeit less efficiently. In human patients, point mutations in ninein have been linked to Seckel syndrome, manifested by dwarfism and microcephaly, in addition to skeletal dysplasia (*Dauber et al., 2012*; *Grosch et al., 2013*). While the mechanisms leading to dysplasias in these patients remain unknown, it is tempting to speculate whether ossification defects due to a reduced pool of multinucleated osteoclasts are at the origin of these abnormalities.

## Materials and methods

### Mice

Ninein-deficient mice were generated as previously described (*Lecland et al., 2019*). Ninein-deleted animals were viable and able to reproduce. Heterozygous mice for the recombined deleted ninein allele (del/+) which behave as wild type were intercrossed and the progeny was characterized following genotyping. Control mice were either wild-type or heterozygous mice, as specified in the figure legends. Homozygous ninein deleted mice for both alleles were mentioned as del/del. Mice expressing the *lacZ* reporter gene in the targeted *Nin* locus (*Nin^{tm1a}*) were generated in the animal facility as already published (*Lecland et al., 2019*) and used to clarify the expression pattern of ninein during embryonic development. For osteoclast ex vivo experiments, we used mice carrying the floxed ninein allele which was previously shown to be functional and displaying a wild type phenotype (*Lecland et al., 2019*). All animal experiments were approved by the Institutional Animal Care and Use Committee at the Genotoul Anexplo facilities of the Center for Integrative Biology, University Toulouse III (institution agreement #D3155511, project agreement APAFIS#2725–2015111213203624 v5).

### Skeleton preparations

Following euthanasia, whole embryos at embryonic day E16.0, E16.5, and E18.5 and neonates at 21 days post coitum were collected, washed in PBS and fixed in 95% ethanol. Skin and internal organs were then removed. Specimens were first stained for cartilage and incubated in a solution of 0.03% Alcian Blue and 20% acetic acid in 70% ethanol for a week at room temperature, under gentle motion. After rinsing in 96% ethanol and incubation in 1% KOH for 1 day, specimens were further stained for bone overnight with 0.01% Alizarin Red in 1% KOH and cleared in 1% KOH. To gain optimal transparency, samples were transferred into a solution of KOH 0.5%/glycerol (vol/vol) for few days and stored at 4 °C in ethanol/glycerol (vol/vol) containing 0.2% NaN$_3$. To perform long bone measurement in forelimb and hindlimb, scapula, humerus, radius, ulna, femur, tibia, and fibula were microdissected in both control and ninein-deleted E16.5 and E18.5 embryos. Samples were measured using Leica MZFLIII stereomicroscope with Leica Application Suite software.

### Histology

Whole embryos at embryonic day E14.5, E16.5, and E18.5 were processed for routine histology following fixation with 4% paraformaldehyde in PBS, dehydration through a series of graded ethanol, clearing in xylene and embedding in paraffin. Additional isolated forelimbs, hindlimbs and E16.5 heads were collected and treated as well. Serial sectioning was performed at 5 μm thickness. Standard staining procedures such as Goldner's trichrome, picrosirius red, toluidine blue, safranin O, and pentachrome method were used to characterize bone and joints development according to published protocols (*Doello, 2014*; *Schmitz et al., 2010*). Following von Kossa-staining to demonstrate bone mineralization, osteoblasts and osteoclasts were enzymatically detected by alkaline phosphatase (AP) or tartrate resistant acid phosphatase (TRAP) staining, respectively, using standard procedures. To assess bone mineralization, von-Kossa-stained sections were analyzed using Fiji software (*Schindelin et al., 2012*). Briefly, von-Kossa-positive areas (brown to black deposits) were measured following thresholding and binarization, and expressed as percentages of the areas of interest. Osteoblasts were quantified manually by counting cuboidal cells located along collagenous matrix, on toluidine-blue-stained sections of E16.5 and E18.5 tibiae.

Ninein expression at early stages of bone development was monitored in E16.5 *Nintm1a* embryos (*Lecland et al., 2019*). Embryos were collected and fixed for 2 hr in 0.2% glutaraldehyde in PBS containing 2 mM MgCl$_2$ and 5 mM EGTA. Some embryos were then longitudinally halved and had their skin removed and fixed for 2 additional hours at 4 °C. Following rinses in PBS, samples were further rinsed in PBS containing ionic (0.01% sodium deoxycholate) and non-ionic (0.02% NP-40) detergents. Overnight incubation was performed at 37 °C with 0.5 mg/ml X-Gal substrate in PBS buffer containing 5 mM of potassium ferro- and ferri-cyanide. Samples were then fixed and embedded in paraffin. Sections were stained with alizarin red. Acquisition of histology images was performed following automatic scanning using Hamamatsu Nanozoomer HT with a 20 x objective or using Nikon Eclipse 80i microscope with a 40x1.4 NA objective, Nikon DMX1200 camera and NIS Elements AR software.

## X-ray microtomography

Tibiae from E18.5 control and ninein-deleted embryos were submitted to X-ray computed microtomography to evaluate the bone morphology. A Phoenix/GE Nanotom 180 instrument using a diamond target (mode 0) was used at a voltage of 70 kV and a current of 300 µA. Samples were positioned at 11 mm from the RX target and at 400 mm from the detector, with a counting time of 750ms per picture and an average of five pictures per 0.25°step. Datos X software was used to process the data and reconstruct 3D images of the bones. Images were treated using Vg Studio Max software. The maximum voxel size was 1.4 µm.

## Ex vivo characterization of osteoclasts

Bone marrow cells were isolated following mechanical disaggregation of long bones from P0 mice, 4 hr after birth. Briefly, after dissection of both forelimb and hindlimb in PBS, bones were freed of muscles and tendons with scalpel and forceps and placed in serum-free α-MEM. Both ends of all long bones were discarded. All central vascularized and bony regions were then longitudinally cut and curetted with a scalpel blade to release cells of the marrow-filled diaphyses into the medium. After pipetting up and down and filtering through a 40 µm cell strainer, cells were centrifuged at 330 × *g* for 5 min, seeded onto glass coverslips in a 24-well plate, and allowed to attach. Cells were grown for 1 hr at 37 °C with 5% CO$_2$ in serum-free α-MEM. Adhesive cells were then fixed with 4% paraformaldehyde in PBS at room temperature for 30 min to perform immunohistochemistry and TRAP staining. All TRAP-positive cells were counted and classified as mononucleate cells, multinucleate osteoclasts, cells within a fusion cluster, or part of an aggregate of unfused cells.

## Bone-marrow-derived osteoclasts and microscopic techniques

For immunofluorescence, coverslips were fixed in methanol at –20 °C for 10 min, followed by staining with primary antibodies against alpha-tubulin (monoclonal DM1A, Sigma-Aldrich), gamma-tubulin (rabbit serum R75, *Julian et al., 1993*), centrin (monoclonal 20H5, Sigma-Aldrich), dynactin p150glued (cat. no 612708, BD Transduction Laboratories), and ninein (rabbit serum 1732, *Srsen et al., 2009*). Images were acquired on a Zeiss Axiovert 200 M inverted microscope using a 100 x/1.4NA objective and an AxioCam MRm camera. Osteoclast cultures from adult bone marrow cells and bone resorption assays were prepared as described (*Vérollet et al., 2013*), in αMEM medium, containing 10% fetal calf serum, 1% glutamine, 30 ng/ml mouse M-CSF, and 50 ng/ml RANK-L (Miltenyi Biotec). Bone marrow cells were cultured in the absence of RANK-L for 2 days prior to adding complete medium, or exposed directly to complete medium upon harvest (in *Figure 7C and D*, *Figure 7—figure supplement 1C–G*). Measurements of the cell surface and the number of nuclei of syncytial osteoclasts were performed on coverslips of TRAP-stained cells. Using a 10 x lens with phase contrast, neighboring fields of view of the entire coverslip were recorded and subsequently analyzed manually with ImageJ, using the 'freehand' selection tool to outline the cell borders, followed by the 'measurement' function. Nuclei were visible by phase contrast and counted manually for each syncytium. Alternatively, for the measurement of the osteoclast fusion index, paraformaldehyde-fixed coverslips were stained with phalloidin-Alexa 488 and 4',6-diamidino-2-phenylindole (DAPI), to visualize actin along the podosome belt, and nuclei respectively. Using ImageJ 'freehand' selection along the podosome belts as well as a home-written macro, the number of nuclei in syncytia was determined relative to the total number of nuclei in a field of view. Each field of view had a surface of 1,700,000 µm$^2$.

Cell adhesion to 96-well culture plates (100,000 cells/well) was quantified after fixation with 4% paraformaldehyde in PBS, rinsing, and staining with 0.1% crystal violet for 30 min, followed by rinsing in water and solubilization of the dye in 2% sodium dodecyl sulfate for 30 min. Absorption of the dye was measured at 550 nm.

Centriole clustering in osteoclasts was examined in methanol-fixed cultures after 3 days of RANK-L treatment, following immunofluorescence of γ-tubulin. Quantification of clusters was performed as described in *Phan et al., 2004*.

## Flow cytometry

Cells were harvested with Accutase (Sigma-Aldrich), rinsed with PBS, and collected by centrifugation at 500 x $g$ for 5 min, then stained with the LIVE/DEAD kit (Fisher Scientific) according to the manufacturer's instructions. Cells were counted and seeded into a 96-well plate at $3\times10^5$ cells/well, stained with 100 µl fluorescently labelled antibodies (anti β1, β2, β3-integrin, Biolegend) for 30 min at 4 °C, then washed twice in PBS. For intracellular staining, cells were fixed and permeabilized using the Foxp3/Transcription Factor Staining Buffer Set (eBioscience) according to the manufacturer's instructions, and stained with 100 µl of Ki-67 antibody (Biolegend) for 30 min at 4 °C, then rinsed twice with PBS.

For cell cycle experiments, $1\times10^6$ cells were collected and fixed in ice-cold 70% ethanol for 30 min at 4 °C. Cells were then stained with propidium iodide (Thermo Fisher) for 15 min. All data were acquired using a BD Fortessa X20 flow cytometer (BD Biosciences), driven by BD FACS Diva software, and analysed using FlowJo (Tree Star, USA).

## Western blot analysis

Cells were detached with accutase and lysed in 50 mM HEPES, pH 7.5, 150 mM NaCl, 1 mM $MgCl_2$, 1 mM EGTA, 0.5% Triton X-100, 1 mM dithiothreitol, and protease inhibitors. Forty µg of extracts were separated on a 12.5% polyacrylamide SDS gel and transferred to a nitrocellulose membrane. Blots were probed with antibodies against cleaved caspase 3 (Cell Signaling Technology, #9664), PARP (Cell Signaling Technology, #9542), and GAPDH (Santa Cruz Biotechnology, #32233), followed by horse-radish-peroxidase-coupled secondary antibodies. Chemiluminescence signal was detected using Amersham ECL Prime Western Blotting Detection Reagent (GE Healthcare, Chicago, IL, USA) on a ChemiDoc Touch imaging system (Biorad).

## Statistical analyses

Statistical analysis was performed using Microsoft Excel and GraphPad Prism 9 (GraphPad Software Inc). Data were expressed as means ± standard deviation, and unpaired Student's t-tests were applied on data sets with a normal distribution (determined using a Shapiro-Wilk test), unless indicated otherwise in the figure legends. $p \leq 0.05$ was considered as the level of statistical significance (*, $p \leq 0.05$; **, $p \leq 0.01$; ns, not significant). For assays using primary cells, experiments were repeated independently four times and representative data were shown.

## Acknowledgements

We are indebted to the Animal Facility at the CBI-FR3743/Université Paul Sabatier Toulouse III for providing the strains used in this study. We thank F Capilla for the use of the histology facilities at the Centre Regional d'Exploration Fonctionnelle et de Ressources Expérimentales, Toulouse Purpan, A Le Ru and C Pouzet from the Toulouse Reseau Imagerie and FR Agrobiosciences Interactions Biodiversity for automated light microscopy imaging (Nanozoomer Hamamatsu) and C Rouvière for help with bone morphometry analysis using Image J software. The French FERMaT Federation FR3089 is acknowledged for providing X-ray tomography laboratory facilities. This work was funded in part by a donation to the University Toulouse III ("Financement maladies orphelines", Simone & Narcisse Bernard), and by grants awarded to CV's team from Agence Nationale de la Recherche (ANR16-CE13-0005-01) and Fondation pour la Recherche Médicale. Further support was provided by Centre National de la Recherche Scientifique, Université Toulouse III, and Institut National de la Santé et de la Recherche Médicale. CG, MP, and OD received doctoral scholarships from Université Toulouse III, and additional support from Fondation ARC pour la recherche contre le cancer (OD), and from the CARE graduate school (MP).

## Additional information

### Funding

| Funder | Grant reference number | Author |
|---|---|---|
| Agence Nationale de la Recherche | ANR16-CE13-0005-01 | Christel Vérollet |
| Universite Toulouse III | Financement S&N Bernard | Andreas Merdes |

The funders had no role in study design, data collection and interpretation, or the decision to submit the work for publication.

### Author contributions

Thierry Gilbert, Conceptualization, Data curation, Formal analysis, Investigation, Visualization, Methodology, Writing - original draft, Writing - review and editing; Camille Gorlt, Marianna Plozza, Laurence Haren, Data curation, Investigation, Writing - review and editing; Merlin Barbier, Benjamin Duployer, Ophélie Dufrancais, Laure-Elene Martet, Elisa Dalbard, Loelia Segot, Investigation; Christophe Tenailleau, Data curation; Christel Vérollet, Funding acquisition, Writing - review and editing; Christiane Bierkamp, Conceptualization, Investigation, Methodology, Writing - review and editing; Andreas Merdes, Conceptualization, Data curation, Supervision, Funding acquisition, Writing - review and editing

### Author ORCIDs

Christel Vérollet (i) http://orcid.org/0000-0002-1079-9085
Andreas Merdes (i) http://orcid.org/0000-0002-3739-2728

### Ethics

All animal experiments were approved by the Institutional Animal Care and Use Committee at the Genotoul Anexplo facilities of the Center for Integrative Biology, University Toulouse III (institution agreement #D3155511, project agreement APAFIS#2725-2015111213203624 v5).

### Decision letter and Author response

Decision letter https://doi.org/10.7554/eLife.93457.sa1
Author response https://doi.org/10.7554/eLife.93457.sa2

## Additional files

### Supplementary files

• Supplementary file 1. Forelimb and hindlimb bone characteristics in control and ninein-deleted mice. Table, indicating the lengths of various bones from embryos at E16.5 and E18.5, from control and ninein del/del embryos.

• MDAR checklist

### Data availability

Original data has been deposited at https://doi.org/10.6084/m9.figshare.25650942. Source data files have been provided for *Figure 7—figure supplement 1*.

The following dataset was generated:

| Author(s) | Year | Dataset title | Dataset URL | Database and Identifier |
|---|---|---|---|---|
| Gilbert T | 2024 | Loss of ninein interferes with osteoclast formation and causes premature ossification | https://figshare.com/articles/figure/original_data_Gilbert_et_al_revised_manuscript/25650942 | figshare, 10.6084/m9.figshare.25650942 |

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
