## [Editor Report]

This valuable study offers new insight into the role of centrosome protein ninein in skeletal development through an analysis of the skeletal phenotype of ninein-deficient mice. The evidence supporting the conclusion is convincing. This work will be of interest to scientists in bone biology and skeletal development field.

---

## [Decision Letter]

**Decision letter after peer review:**

Thank you for submitting your article "Loss of ninein interferes with osteoclast formation and causes premature ossification" for consideration by *eLife*. Your article has been reviewed by 3 peer reviewers, and the evaluation has been overseen by a Reviewing Editor and Kathryn Cheah as the Senior Editor.

In this study, the authors analyzed the skeletal phenotype of ninein-deficient mice and investigated the role of this centrosomal protein in regulating osteoclast activities. Reviewers have raised concerns, primarily regarding the descriptive nature of the paper. There is insufficient exploration of how reduced ninein causes bone defects and how ninein-related centrosome function regulates osteoclast activities. The authors should also address numerous other concerns raised by the reviewers.

*Reviewer #2 (Recommendations for the authors):*

1) Although the findings in Figure 1 on ninein del/del females and embryos are very interesting and worth reporting, as the authors mention; in my opinion, they do not contribute and are not relevant to the described bone phenotypes resulting from ninein del/+ crosses.

2) In this respect, perhaps the rationale for including the data above is that the authors analyzed homozygous del/del "dead" pups at 22 days post-coitum in Supp Figure 3. However, how can the authors be sure that the wild-type "neonates are indeed "time-matched" as they mention? A difference in a few hours or one day can conceivably result in more ossifications in the mutants.

3) Quantifications and statistical analyses in Figure 7D-G are necessary to make the correlations that the authors claim in the manuscript.

*Reviewer #3 (Recommendations for the authors):*

1. Writing: The authors put much emphasis on the centrosome in the Introduction session. However, it was not until Figure 7 did they show abnormal centriole clustering in osteoclasts. The introduction should include more background on osteoclast and osteoblast balance during skeletal development.

2. In figure 2C, it should explain what "*" mean.

3. In figure 3G-H, the images seem to be not equally magnified.

[Editors' note: further revisions were suggested prior to acceptance, as described below.]

Thank you for resubmitting your work entitled "Loss of ninein interferes with osteoclast formation and causes premature ossification" for further consideration by *eLife*. Your revised article has been evaluated by Kathryn Cheah (Senior Editor) and a Reviewing Editor.

The manuscript has been improved but there is a remaining issue that needs to be addressed, as suggested by Reviewer #2.

*Reviewer #2 (Recommendations for the authors):*

For the earlier phenotypes shown in Figure 1, the authors may want to add to the discussion the possibility that ninein may play a similar role in centrosome clustering of other cell types, for example the multi-nucleated trophoblast giant cells, similar to that of osteoclasts, leading to embryonic arrest or developmental delay (PMID: 27902976, PMID: 36001376).

---

## [Author Response]

In this study, the authors analyzed the skeletal phenotype of ninein-deficient mice and investigated the role of this centrosomal protein in regulating osteoclast activities. Reviewers have raised concerns, primarily regarding the descriptive nature of the paper. There is insufficient exploration of how reduced ninein causes bone defects and how ninein-related centrosome function regulates osteoclast activities. The authors should also address numerous other concerns raised by the reviewers.Reviewer #2 (Recommendations for the authors):1) Although the findings in Figure 1 on ninein del/del females and embryos are very interesting and worth reporting, as the authors mention; in my opinion, they do not contribute and are not relevant to the described bone phenotypes resulting from ninein del/+ crosses.2) In this respect, perhaps the rationale for including the data above is that the authors analyzed homozygous del/del "dead" pups at 22 days post-coitum in Supp Figure 3. However, how can the authors be sure that the wild-type "neonates are indeed "time-matched" as they mention? A difference in a few hours or one day can conceivably result in more ossifications in the mutants.

Mating of mice was performed overnight, and there is indeed an uncertainty of a few hours concerning the exact time post-coitum. Nevertheless, similar sizes of pups of the two genotypes at 22 dpc allowed us to conclude that any difference should be minimal.

In most experiments (Figures 2, 3, 4, 5, 6), we eliminated this uncertainty by comparing heterozygous controls to homozygous del/del embryos from the same litter (from crossings between +/- heterozygous parents), thus achieving perfect time match. DNA samples from each embryo were analyzed by genotyping.

3) Quantifications and statistical analyses in Figure 7D-G are necessary to make the correlations that the authors claim in the manuscript.

To evaluate the role of ninein in centrosome clustering, quantifications were performed and included in the new Figure 8H, as well as on page 10 of the revised manuscript. A significant defect in centrosome clustering is visible for ninein del/del osteoclasts.

Reviewer #3 (Recommendations for the authors):1. Writing: The authors put much emphasis on the centrosome in the Introduction session. However, it was not until Figure 7 did they show abnormal centriole clustering in osteoclasts. The introduction should include more background on osteoclast and osteoblast balance during skeletal development.

To address this, we included more background on the role of osteoclasts and osteoblasts in the revised introduction (page 4).

2. In figure 2C, it should explain what "*" mean.

We included this information in the legend to Figure 2C, on page 25 of the revised manuscript (“… more intense bone staining…”). Moreover, details on how statistical analyses were performed can now be found in the methods section of the revised manuscript.

3. In figure 3G-H, the images seem to be not equally magnified.

We have corrected this in the revised figure (now Figure 3E, F).

[Editors’ note: what follows is the authors’ response to the second round of review.]

The manuscript has been improved but there is a remaining issue that needs to be addressed, as suggested by Reviewer #2.Reviewer #2 (Recommendations for the authors):For the earlier phenotypes shown in Figure 1, the authors may want to add to the discussion the possibility that ninein may play a similar role in centrosome clustering of other cell types, for example the multi-nucleated trophoblast giant cells, similar to that of osteoclasts, leading to embryonic arrest or developmental delay (PMID: 27902976, PMID: 36001376).

In response to your letter, we have addressed the comments of reviewer 2 and have added an additional paragraph to our discussion (track changes on page 13, top, and page 14, end of first paragraph). We discuss the potential effect of centrosome clustering in trophoblast giant cells on embryonic development, and we cite the two related references.